# Preparation and Characterization of Plasma-Derived Fibrin Hydrogels Modified by Alginate di-Aldehyde

**DOI:** 10.3390/ijms23084296

**Published:** 2022-04-13

**Authors:** Raúl Sanz-Horta, Ana Matesanz, José Luis Jorcano, Diego Velasco, Pablo Acedo, Alberto Gallardo, Helmut Reinecke, Carlos Elvira

**Affiliations:** 1Department of Applied Macromolecular Chemistry, Institute of Polymer Science and Technology, Spanish National Research Council (ICTP-CSIC), Juan de la Cierva 3, 28006 Madrid, Spain; gallardo@ictp.csic.es (A.G.); hreinecke@ictp.csic.es (H.R.); celvira@ictp.csic.es (C.E.); 2Department of Bioengineering and Aerospace Engineering, Universidad Carlos III de Madrid (UC3M), 28911 Madrid, Spain; amatesan@pa.uc3m.es (A.M.); jjorcano@ing.uc3m.es (J.L.J.); divelasc@ing.uc3m.es (D.V.); 3Department of Electronic Technology, Universidad Carlos III de Madrid (UC3M), 28911 Madrid, Spain; pag@ing.uc3m.es; 4Instituto de Investigación Sanitaria Gregorio Marañón, 28007 Madrid, Spain

**Keywords:** fibrin hydrogels, plasma-derived fibrin hydrogel, platelet-poor plasma (PPP), oxidized alginate, alginate di-aldehyde hydrogels, tissue engineering, skin tissue engineering, skeletal muscle regeneration, bone tissue reparation

## Abstract

Fibrin hydrogels are one of the most popular scaffolds used in tissue engineering due to their excellent biological properties. Special attention should be paid to the use of human plasma-derived fibrin hydrogels as a 3D scaffold in the production of autologous skin grafts, skeletal muscle regeneration and bone tissue repair. However, mechanical weakness and rapid degradation, which causes plasma-derived fibrin matrices to shrink significantly, prompted us to improve their stability. In our study, plasma-derived fibrin was chemically bonded to oxidized alginate (alginate di-aldehyde, ADA) at 10%, 20%, 50% and 80% oxidation, by Schiff base formation, to produce natural hydrogels for tissue engineering applications. First, gelling time studies showed that the degree of ADA oxidation inhibits fibrin polymerization, which we associate with fiber increment and decreased fiber density; moreover, the storage modulus increased when increasing the final volume of CaCl_2_ (1% *w/v*) from 80 µL to 200 µL per milliliter of hydrogel. The contraction was similar in matrices with and without human primary fibroblasts (hFBs). In addition, proliferation studies with encapsulated hFBs showed an increment in cell viability in hydrogels with ADA at 10% oxidation at days 1 and 3 with 80 µL of CaCl_2_; by increasing this compound (CaCl_2_), the proliferation does not significantly increase until day 7. In the presence of 10% alginate oxidation, the proliferation results are similar to the control, in contrast to the sample with 20% oxidation whose proliferation decreases. Finally, the viability studies showed that the hFB morphology was maintained regardless of the degree of oxidation used; however, the quantity of CaCl_2_ influences the spread of the hFBs.

## 1. Introduction

Tissue engineering is a field that, making use of scaffolds, cells or growth factors, develops new products and strategies to repair, substitute or generate new organs or damaged tissues [1,2,3]. Hydrogels are one of the most popular scaffolds used in tissue engineering due to their high-water retention capacity, as well as their viscoelastic mechanical properties that mimic human living tissues and organs [4,5]. During the past years, numerous publications have described the synthesis of hydrogels using synthetic polymers because of the physical-chemical control of their properties. However, the use of polymers derived from natural sources demonstrates great properties in regard to biocompatibility and biodegradability [6,7].

On the one hand, alginate is a lineal anionic polysaccharide which is obtained from bacteria and seaweed. Its composition is based on blocks of β-(1-4)-linked D-mannuronic acid (M) and α-(1-4)-linked L-guluronic acid (G), the distribution of which depends on the source [8]. Alginate is a biomaterial commonly used in the food industry, pharmacology and tissue engineering due to its capacity to form physical hydrogels with divalent cationic elements, such as Ca^2+^, Ba^2+^, etc. [9]. In addition, alginate can also form covalent hydrogels through chemical modifications of the carboxylic acid or hydroxyl groups adding new functional groups to react with a crosslinker [8]. Oxidation of alginate by sodium metaperiodate is one of the best known chemical modifications to obtain aldehyde groups in the polysaccharide structure [10]. The sodium metaperiodate proceeds by oxidizing hydroxyl groups from the C-2 and C-3 positions of an alginate monomeric repeating unit, which results in an opening of the ring that contains two aldehyde groups that may react with primary amines, leading to Schiff base formation [11].

On the other hand, fibrin is a biopolymer implicated in the blood coagulation process for natural wound healing. Fibrin is formed due to fibrinogen polymerization by thrombin or CaCl_2_, leading to the formation of a 3D network of fibrin fibers. Similar to collagen hydrogels, fibrin-based hydrogels facilitate a high efficiency of cell seeding and uniform cell distribution by proliferating, migrating, and differentiating into specific tissues/organs through the secretion of the extracellular matrix (ECM), while resorbing gradually due to the action of proteases [12,13]. Using these physiological properties together with cell-material interactions, the fibrin matrices have been utilized in tissue engineering applications such as skin tissue, adipose tissue, ocular tissue, cardiovascular tissue, blood vessels, musculoskeletal tissue, bone tissue or nerve tissue [12,14,15,16,17]. Special attention should be paid to the use of human plasma-derived fibrin hydrogels as a 3D scaffold in the production of autologous skin grafts, skeletal muscle regeneration and bone tissue reparation, among other examples [18,19,20,21,22,23,24,25,26,27,28]. In this case, the blood plasma composition (plasma proteins, immunoglobulins, growth factors, enzymes, vitamins, platelets and hormones) provides a more suitable 3D environment to promote migration, proliferation and differentiation of the cells [13,29]. Another crucial component existing in plasma are the platelets, whose concentration influences cell proliferation due to a high content of growth factors and cytokines. Platelet-rich plasma (PRP) has shown the promotion of cell growth in various tissue engineering applications such as bone, cartilage and skin [25,30,31,32,33,34,35,36]. The difficulty of controlling growth factors and cytokine release in PRP has made its combination with other polymers to regulate its leak crucial [33,35,36,37,38,39]. Although PRP has shown its potency and capability in tissue regeneration, its inconsistency between experiments and low reproducibility has made prevalent the use of platelet-poor plasma (PPP) in applications in which platelet-released growth factors are not crucial. Additionally, as PPP is cost-effective, suitable for bulk production and easily translatable with minimal regulatory requirement for FDA approval, it is being widely used in tissue regeneration [20,26,40,41,42]. Despite its versatility, fibrin’s poor mechanical properties, shrinking behavior and fast biodegradation limit its applicability in cell culture [43,44]. During the last years, the combination of fibrin with natural or synthetic polymers is being investigated in order to overcome the aforementioned disadvantages [11,12,13,45,46,47,48]. In particular, there are some authors describing the combination of fibrin-alginate as an interpenetrated network to enhance original fibrin properties for in vitro growth of ovarian follicles [49], as a wound healing hydrogel [50] and for the preparation of microbeads based on alginate-fibrin or alginate di-aldehyde (ADA)-fibrin for controlled released of stem cells [51].

Herein, plasma-derived fibrin and alginate di-aldehyde were combined to produce natural hydrogels for tissue engineering applications. A similar study on oxidized alginate in combination with fibrin has been described [52]. In comparison, in our work, we have used platelet-poor plasma with low concentrations of fibrinogen to prepare the hydrogels by adding low concentrations of oxidized alginates, using different ranges of ADA modifications, while also studying the influence of CaCl_2_ in fibrin polymerization and on the crosslinking of the prepared hydrogels. In addition, changes in behavior and mechanical properties of hydrogels were studied in order to elucidate how alginate di-aldehyde impacts fibrin hydrogel formation. The microstructure of fibrin-ADA gels were analyzed by Scanning Electron Microscopy (SEM) in order to observe the relationship between the structure and the behavior of the hydrogels. Knowing that fibroblasts are the characteristic and most abundant cell type of the connective tissue, primary human fibroblasts (hFBs) were seeded inside fibrin-alginate hydrogels in order to evaluate their biocompatibility. Proliferation studies were performed at different time points, 1, 3, and 7 days, and viability essays were performed at 3 and 7 days.

## 2. Results and Discussion

### 2.1. Alginate-di-Aldehyde Preparation and Characterization

Sodium-alginate was oxidized by NaIO_4_ in aqueous solution to give alginate di-aldehyde. The percentage of alginate oxidation was measured according to the procedure developed by Zhao and Heindel [53], obtaining the modification shown in Table 1.

Figure 1 shows the ^1^H-NMR spectra of sodium alginate and different samples of alginate-di-aldehyde in D_2_O. For sodium alginate, the signals at 4.95 and 4.13 ppm correspond to H1-G and H3-G, respectively.

For alginate di-aldehyde, the signals tending to spread when increasing the theoretical oxidation percentage could be associated with alginate cleavage during the oxidation reaction or to inter and intramolecular hemiacetal formation due to ring opening oxidation. As hemiacetal groups form because of the reaction between aldehyde and neighbor hydroxyl groups, new signals at 5.6 and 5.3 ppm appear due to the presence of hemiacetal protons [54,55]. As expected, the hemiacetal proton signals increase for higher oxidation percentages. The signal at 3.55 ppm corresponds to the remaining ethylene glycol added to stop the reaction.

Figure 2a shows the FTIR (Fourier transform infrared spectroscopy) spectra of sodium alginate, dry ADA80 and wet ADA80 that was left in a closed chamber at 100% relative humidity for 2 h before measurement. As can be seen in all three cases, a broader band was observed at 3600–2800 cm^−1^, related to the OH stretching vibration being more pronounced for wet ADA80 due to moisture in the sample. All samples show a weak band at 2900 cm^−1^, associated with the CH stretching vibration. Typical C=O carboxylate signals appear for all samples at 1593 cm^−1^ as an antisymmetric stretch, and at 1407 cm^−1^ as a symmetric stretch.

In the spectra of the wet ADA80 sample, the antisymmetric stretch signal shifted to 1610 cm^−1^ and broadened due to the presence of water in the sample. Interestingly, a new band in the wet ADA80 sample appears at 1723 cm^−1^ that is directly related to aldehyde groups as shown in reference [56]. This new band only appears in wet samples due to the existing equilibrium between hemiacetal groups and aldehyde groups. For this reason, the equilibrium was shifted towards aldehyde formation by moistening the samples as shown in Figure 2a. Another signal to obtain this equilibrium shift was observed at 880 cm^−1^, where the band associated with hemiacetal groups disappears in the wet ADA80 sample [57].

In order to prove that ADA samples are able to react with aminated products, ADA samples were dissolved in water to react with an excess of mono-Jeffamine, with the intention of observing changes in the peak intensities of the dry products. A Schiff base characteristic signal appears at 1639 cm^−1^ [55]. The spectra were normalized with respect to the band at 1639 cm^−1^ and the band at 1100 cm^−1^, associated with the C-O stretch vibration of mono-Jeffamine, was analyzed. It was observed that the intensity of the C-O band of the mono-Jeffamine varies in correlation with the oxidation degree of each ADA sample.

TGA of sodium alginate shows typical water loss (14%-wt) and a peak at 245 °C in the first derivative of the weight loss curve (31.7% weight loss) that is associated with carboxylate thermal cleavage and depolymerization leading to CO_2_ formation [58]. TGA thermograms of alginate di-aldehyde are slightly different from that of sodium alginate, as new peaks in the first derivative of the weight loss curve are present that may be associated with polymer chains of different molecular weights. However, the peak at 245 °C, associated with thermal cleavage of carboxylate groups, remains for every alginate di-aldehyde sample. These TGA results are in line with the molecular weights obtained by GPC from different alginate di-aldehydes, as shown in Table 1.

### 2.2. Gelation Time and Kinetics of Fibrin/ADA Gels

ADA with different degrees of oxidation was incorporated in fibrin precursor solutions in order to modify its hydrogel properties. Firstly, the maximum amount of ADA of each sort that allows for fibrin polymerization and gel formation was analysed by timing gelation of fibrin by the flip-flop method. As can be seen in Table 1, the maximum amount of ADA for gel formation was 3 mg/mL and the gelation time was delayed until ADA50 eventually inhibited formation of a fibrin hydrogel. Gelling time was analysed by UV spectroscopy, in which the absorbance of the solution was monitored at 350 nm, in order to have a closer look at how fibrin polymerization evolves. Figure 3 depicts the gelling time delay as a function of the amount of ADA and its modification increase due to the aforementioned inhibition of fibrin polymerization. Complete polymerization inhibition was observed for ADA80. This inhibitor effect could be attributed to a fibrinogen coating by the alginate di-aldehyde in which alginate is more likely to bind fibrinogen, with a higher degree of modification preventing fibrin monomers to form oligomers between one another as happens in the case of polyphosphate [59]. In addition, ADA can anchor proteins and other biomolecules present in a plasma solution, preventing fibrin monomers from participating in the polymerization. As shown in Figure 2, it seems that fibrin-ADA10 gels tend to be more turbid because the absorbance is higher, which is directly related to the fibres’ size and density, as will be analysed later. As ADA50 and ADA80 were unable to form gels, they were discarded for further experiments.

As alginate is able to form physical gels in the presence of CaCl_2_, adding ADA to a fibrin precursor solution could reduce the amount of Ca^2+^ ions participating in triggering fibrin polymerization. Raising the CaCl_2_ concentration to trigger gelation seems to reduce this inhibitory effect on fibrin polymerization in which ADA10 (200 µL CaCl_2_) has a shorter gelling time compared to that of the fibrin control gel (200 µL CaCl_2_). As explained previously, for ADA20 (200 µL CaCl_2_), the inhibition effect seems to still occur. Cloudier gels were obtained when increasing the CaCl_2_ concentration as this precursor molecule seems to increase the lateral aggregation of fibrin fibres leading to more opaque gels, as explained by Ryan et al. [60].

### 2.3. Microstructure of Fibrin/ADA Hydrogels

The microstructure of the hydrogels was analysed by SEM, submitting gels to a supercritical CO_2_ drying process prior to observation in the microscope. A porous hydrogel structure was expected as alginate-di-aldehyde was able to react with proteins present in human plasma, or to directly react with fibrinogen molecules changing the original structure. However, when looking at the microstructure of hydrogels (see Figure 4), a messy fibre network characteristic of fibrin hydrogels was observed for each sample. Nevertheless, by measuring fibres, a tendency to increase the fibre diameter and reduce the fibre density were noted when adding ADA10 and ADA20, as shown in Figure 4. This slight change in the microstructure could be related to the aforementioned tendency of ADA to inhibit fibrin polymerization.

It is well known that the microstructure of fibrin hydrogels can be varied by using different concentrations of its precursors (fibrinogen, CaCl_2_ and thrombin). As is shown in Figure 4, increasing CaCl_2_ concentrations to start fibrin polymerization promotes fibre aggregation, leading to wider fibres and heterogeneous fibrin networks. In order to observe a stronger contribution of ADA samples to fibrin polymerization, the CaCl_2_ concentration was raised to trigger gelation. For the control fibrin, gelled with 200 µL of 1% of CaCl_2_ solution, fibres became wider and aggregate, leading to a less dense gel with more open holes. It has been shown that adding ADA10 and ADA20, gelling time delays due to polymerization inhibition and the microstructure of gels with higher concentrations of CaCl_2_ show a tendency for fibre aggregation. When adding ADA10 and ADA20 using higher CaCl_2_ concentrations gels with an aggregate microstructure were formed and fibres cannot be distinguished due to higher fibre aggregation. For ADA20 (200 µL CaCl_2_), a densely aggregated gel microstructure was observed. This microstructure of ADA10 and ADA20 (200 µL CaCl_2_) shows a tendency for fibrin monomers to aggregate instead of forming large fibres, leading to more randomized network microstructures, as can be observed in Figure 4e,f.

### 2.4. Rheological Characterization

Rheological tests did not show significant differences when adding ADA to the formulation of a fibrin hydrogel. However, when increasing the concentration of CaCl_2_ in the hydrogels, the storage modulus was raised in the case of ADA10. As alginate di-aldehyde is able to capture Ca^2+^ ions, this could play an important role in fibre formation and the hydrogel’s mechanical properties. A sodium alginate-fibrin hydrogel was gelled to corroborate the role of Ca^2+^ ions on fibrin polymerization, showing a loss in the mechanical properties of the hydrogel in comparison with fibrin gel. In order to observe higher differences between mechanical properties, a variation of CaCl_2_ concentration for starting gelation of fibrin hydrogel was conducted. As observed, fibres tend to aggregate themselves, leading to wider and stronger fibres when increasing the CaCl_2_ concentrations, an aspect that has a strong influence on the mechanical properties and the gel behaviour. Therefore, in order to observe a real ADA contribution to fibrin gel mechanical properties, the CaCl_2_ volume added to form the hydrogel was changed from 80 µL to 200 µL per mL of hydrogel and submitted to rheological characterization. It was observed that for ADA 10 with 200 µL CaCl_2_, the storage modulus raises to 23 Pa; meanwhile, for the fibrin hydrogel with CaCl_2_, the modulus raises to 11 Pa. The thickening fibre effect of CaCl_2_ seems to increase the mechanical properties of fibrin hydrogels, but when adding ADA10 this effect seems to be multiplied. However, when adding ADA20, the mechanical properties of the gels decrease due to the inhibition effect of the polymerization, leading to soft gels.

The microstructure and rheological analysis indicate the influence of adding alginate di-aldehyde to fibrin polymerization leading to more aggregation of fibrin oligomers and inducing changes in the original fibrin gels. Furthermore, the degree of oxidation of the alginates plays an important role on fibrin polymerization inhibition, and the CaCl_2_ concentration can be significant when using alginate to hybridize fibrin hydrogels. In this sense, Vorwald et al. combined alginate and fibrin, adjusting its properties by controlling the addition of thrombin to begin fibrin polymerization, and CaCl_2_ to control alginate crosslinking [61].

Although the storage modulus of the oxidized alginate hydrogels is very low, around ~10–40 Pa (Figure 5), they allow fibroblast adherence and subjection forces in the hydrogel, required for cell spreading. These results agree with previous reports: Sun, Y. et al. showed a rheological characterization of fibrin hydrogels with storage modulus between 1 and 100 Pa [62]. Montero et al. showed cell viability and spreading of cells in this range of storage modulus values [43]. Conversely, the developed 3D-bioplotted hydrogel vessel-like constructs made from ADA and gelatin (GEL) hydrogels, by Distler et al., showed values of approx. 50 kPa [63]. They increased the mechanical stability of fibroblasts, using CaCl_2_ as crosslinker, giving similar results in hFBs’ adherence and spreading, as the results show. The increase in kPa could avoid the rapid contraction of our hydrogels and might be related to the increase in rigidity, resulting in a more rigid matrix that is too stiff to deform by hFBs.

### 2.5. Fibrin/ADA Hydrogel Contraction

The shrink behavior of fibrin/ADA gels was analyzed by studying the weight loss and area reduction of hydrogels in the absence and presence of hFBs, respectively, as depicted in Figure 6. It was observed that when adding ADA10 and ADA20 to fibrin gels, the shrink behavior deteriorates, as fibrin/ADA gels contract more than the control sample. The microstructure is a key factor of the hydrogel behavior, so the widening fiber effect of ADA also reduces the number of fibers, leading to a lower water-holding capacity of the modified hydrogels. Moreover, for gels polymerized through a higher CaCl_2_ concentration, despite exhibiting denser microstructures, it seems that the water-holding capacity is reduced. For all cases, during the first 8 h, samples present rapid contraction, reaching a sort of plateau after 2 days, but still shrinking.

However, when measuring the area contraction of ADA/fibrin hydrogels in the presence of hFB, this shrinkage effect seems to be slightly reduced. This can be explained due to the shrinkage in the presence of cells not only being related to the water-holding capacity or to the physico-chemical interactions of the fibrin network, but also to the degradation of the fibrin network that plays a vital role in the long-term stability of fibrin hydrogels. As the microstructure of ADA/fibrin hydrogels is characterized by a slight increase in fiber diameter, this could be the reason why, in presence of cells, the area shrinkage of hydrogels is reduced.

### 2.6. Proliferation and Live/Dead Assays: Encapsulated hFBs

Proliferation of primary human hFBs inside the hydrogels was studied through the AlamarBlue TM assay (Figure 7). In general, hFBs embedded in hydrogels with 200 µL of calcium chloride exhibited better behavior in terms of cell proliferation than the hydrogels with 80 µL. In both cases, the percentage of alginate oxidation inside the hydrogels also affects the cell proliferation: in the first days (1 and 3 days), the hydrogels with 80 µL of CaCl*_2_*-oxidized alginates showed fluorescence values higher than those of the controls (only plasma hydrogel). The graph at 7 days also displays that, when the alginate with 10% oxidation is added, the results are similar to those of the control, in contrast to what happens when adding alginates with 20% oxidation. This is in agreement with previous reports on the proliferation of cells. Somasekharan et al. developed hydrogels formed by oxidized alginate (varying the degree of oxidation), PRP and gelatin [64]. Their cell culture studies (L929 cells) suggest that the cells encapsulated in the cell-laden constructs were viable, exhibiting more than 80% cell viability at 24 h. They also show that the viability of cells at this time point increases by adding oxidized alginate. In addition, Sarker et al. also developed gelatin-oxidized alginate hydrogels with encapsulated FB, where the analysis showed that ADA-GEL hydrogels contain more viable cells with intact nuclei and cells in comparison with pure alginates [65]. Moreover, cells were found to be attached and spread after 4 and 7 days of incubation. They demonstrate that the attachment, proliferation, spreading and viability of human dermal hFBs were significantly enhanced on ADA-GEL hydrogels synthesized by covalent crosslinking of ADA and gelatin, as in our studies. A reason for the decrease in proliferation in the hydrogels with higher percentage of oxidation (20%) was explained by Genç et al. [66]. They studied alginate, ADA and ADA-GEL, and their effect on cells (fibroblasts and endothelial cells), and they showed that there was a concentration-dependent effect of the degree of oxidation on cell viability. In ADA hydrogels, there was an extremely strong ROS generation that resulted in a rapid depletion of cellular thiols, leading to rapid necrotic cell death. The authors showed the relationship between oxidative stress-induced intracellular processes and alginate di-aldehyde-based bioinks. In brief, they assessed the event of reactive oxygen species (ROS) as a result of using oxidized alginate with fibroblasts.

Furthermore, the hydrogels with 10% oxidized alginate and a high amount of calcium chloride (200 µL), presented a high cellular proliferation, similar to that of the control during the first 7 days. On the other hand, Vorwald et al. developed a fibrin-alginate interpenetrating network (IPN) hydrogel [61]. They tested the proliferation (AlamarBlue) of their hydrogels with mesenchymal stromal cells (MSCs) and endothelial cells (ECs) and observed that both, at days 7 and 14, have a better viability when the CaCl*_2_* molarity decreases (from 40 mM to 10 mM) in contrast to what is observed (Figure 7) when the increment in CaCl*_2_* (from 7.2 mM to 18 mM) implies an increase in proliferation. In both cases also, the proliferation is worse than that of the control (only fibrin), as in our study.

Further analysis of the effect of hydrogels containing oxidized alginate was performed using the Live/Dead^®^ assay (Figure 8). This assay served to prove whether human primary hFBs can be embedded inside the plasma and the oxidized alginate hydrogels and whether they can survive by spreading and dividing inside the gels.

The results displayed that the hFBs’ morphology is maintained regardless of the degree of oxidation used; however, the quantity of calcium chloride influences the spread of the hFB (Figure 8a,c). When using 200 µL with a dense aggregate gel microstructure, at day 7, the hydrogels still contain some live no-spread cells. In the Live/Dead study, the proliferation was lower in the cases of 20% oxidized alginate compared with that of the plasma controls (Figure 8a,b); and for 10% oxidized alginate, the viability increased with fewer dead cells that are spread more at day 7 (Figure 8a,c).

These results are in line with the fibrin-alginate (FA) IPN developed by Vorwald et al. [61]. They studied the spreading of stromal cells and endothelial cells inside their IPNs with 10 mM and 40 mM of CaCl_2_ as crosslinker and found that the EC-MSC spreading is a function of the alginate crosslinking density and CaCl_2_ concentration. In the case of 10 mM of CaCl_2,_ as it has a lower storage modulus, greater cell spreading was observed when compared to that of the case of 40 mM. Their quantifications using the total area and the circularity of cells supported these observations.

## 3. Materials and Methods

### 3.1. Materials

Alginic acid sodium salt (A2158-100G, from brown algae, viscosity of 2% solution at 25 °C: ~250 cps), N-hydroxylamine chloride (159417), NaOH, methyl orange, NaCl, CaCl_2_ and ethylene glycol (324558) were purchased from Sigma Aldrich (Burlington, MA, USA). Sodium metaperiodate was purchased from Acros Organics (Geel, Belgium). Jeffamine M-3085 was supplied by Huntsman (Houston, TX, USA). Finally, amchafibrin (500 mg) was supplied by Meda Pharma SL (Madrid, Spain).

### 3.2. Synthesis of Alginate di-Aldehyde

Oxidized alginate was synthesized according to Emami et al. [67]. Briefly, 1 g of alginate was dissolved in 100 mL of distilled water. Sodium metaperiodate was added dropwise in a molar ratio of (sodium metaperiodate/alginate repeat unit) 10, 20, 50 and 80 mol%, as specified in Table 2. The mixture was left for reaction during 6 h in the dark under constant stirring and at room temperature. Ethylene glycol was added in excess of the reacting solution for 30 min, and then the product was precipitated in ethanol to stop the oxidation reaction. The precipitate was collected by centrifugation and vacuum dried. Samples were named as ADAX (10, 20, 50 and 80) according to the molar ratio of NaIO_4_ to sodium alginate repeating unit.

### 3.3. Aldehyde Characterization

The characterization of aldehyde groups in the oxidized alginates was performed by hydroxylamine hydrochloride titration and measuring the pH [67]. For the characterization, 0.1 g of each ADA sample was dissolved in 20 mL of 0.25 N-hydroxylamine hydrochloride solution and stirred for 2 h, checking 3–5 solutions per each condition. The solution was previously prepared in 500 mL of distilled water with 8.75 g of hydroxylamine hydrochloride and 3 mL of methyl orange reagent (0.05%), adjusting its pH to 4.

After 2 h, 0.1 M NaOH (whose molarity was previously determined with potassium hydrogen phthalate) was added to the solutions, using a burette, while the pH was measured. A graph of pH was depicted, and their first derivative calculated, where the maximum value of the first derivative indicates the equivalence point of the titration. Finally, the percentage of oxidation of the alginate was extracted, by using the following equation:(1)ADA %=198 gmol of alginate×0.1M NaOH ×VControl−Vsample0.1 g of ADA ×100

Vcontrol is the original volume of NaOH in the burette and the Vsample is the solution existing in the burette in each measured pH point; the value (Vsample − Vcontrol) indicates the consumed amount of sodium chloride used to reach the equivalence point.

### 3.4. Proton Nuclear Magnetic Resonance

^1^H-NMR spectra of sodium alginate and different alginate-dialdehyde were recorded at 25 °C in 1% (*w/v*) D_2_O solutions with a Varian Gemini 400 MHz under standard conditions.

### 3.5. Characterization by FTIR-ATR (Fourier Transform Infrared Spectroscopy-Attenuated Total Reflection)

Changes in the chemical structure were analyzed by recording the FTIR-ATR spectra on a Spectrum One FTIR spectrometer (Perkin Elmer, Waltham, MA, USA) using the same pressure for each sample and an accumulation of 4 runs.

In order to check if alginate di-aldehydes can anchor aminated products, they were dissolved in water and reacted with an excess of Jeffamine^®^ M-3085. Different ADA samples thus obtained were dissolved in water at 1% (*w/v*) and Jeffamine^®^ M-3085 was added in excess (amine/aldehyde ratio of 3) and stirred for 3 h. The resulting products were dialyzed with dialysis tubing of 14 KDa molecular weight cut-off in distilled water for 2 days to eliminate the unreacted Jeffamine, and finally were lyophilized, for further characterization. The spectra obtained were normalized to the peak at 1639 cm^−1^ (Schiff base) in order to observe differences in peak intensities as a function of the aldehyde content for each sample.

### 3.6. Thermogravimetric Analysis (TGA)

Thermogravimetric analysis was carried out using a TA-Q500 Analyser in order to analyze changes in the thermal degradation of the polymers and to observe variations of the polydispersity. Polymers were heated from room temperature to 500 °C at 10 °C/min.

### 3.7. Gel Permeation Chromatography (GPC)

Samples weight-average, and number-average molecular weights (*M*_W_ and Mn, respectively) and polydispersity were measured by gel permeation chromatography (GPC) by using Waters 1515 Isocratic HPLC Pump calibrated with narrow molecular weight distribution pullulan standards. ADA of different oxidation degrees was dissolved at 2 mg/mL and nitrate was used as the mobile phase at a flow rate of 1 mL/min.

### 3.8. hFBs Culture

hFBs were obtained from skin biopsies of healthy donors from the collections of biological samples of human origin registered in the “Registro Nacional de Biobancos para Investigación Biomédica del Instituto de Salud Carlos III”. The cultured medium was Dulbecco’s modified Eagle’s medium (DMEM, Biochrom KG, Cambridge, UK), containing 10% FBS (Fetal Bovine Serum) and 1% of penicillin/streptomycin. Cells were stored at 37 °C and 5% CO_2_ in the incubator, and media was changed every 2 days.

### 3.9. Fibrin/Alginate di-Aldehyde Hydrogel Preparation

Plasma-derived fibrin hydrogels were prepared following the protocol described by Montero et al. [43]. Plasma aliquots of known fibrinogen concentration were thawed in a water bath at 37 °C. In a typical experiment, to prepare 200 µL of plasma hydrogel in a glass vial, 90.91 μL of plasma (with a fibrinogen concentration of 2.64 mg/mL), 16 μL of Amchafibrin, 83.49 μL of saline (NaCl 0.9 % (*w/v*)) and 16 μL of CaCl_2_ (prepared at 1% *w/v* in saline) were sequentially added to the vial and mixed. The final concentrations of fibrinogen and CaCl_2_ in the plasma hydrogels were adjusted to 0.12% and 0.08% (*w/v*), respectively, by conveniently varying the plasma and saline volumes used. To prepare the plasma/ADA hydrogels, the volume of NaCl used in the protocol for the preparation of plasma fibrin-derived hydrogels was replaced with the necessary volume of ADA (predissolved in NaCl 0.9 % (*w/v*)) to reach a final concentration of 0.2% *w/v*. All hydrogels were incubated for 1 h at 37 °C and 5% CO_2_ for complete gelation.

### 3.10. Gelation Time and Kinetics

Gelling times of fibrin/alginate di-aldehyde hydrogels were studied by both flip-flop method and UV spectroscopy to observe the influence of alginate di-aldehyde when incorporated to the plasma solution.

For the flip-flop method, the hydrogel mixtures were prepared and incubated as described above. Vials were then tilted every 1 min. When there was no liquid left in the vial and the hydrogel remained stuck at the bottom, this was considered as the gelation time.

UV spectroscopy was performed using a SynergyTM HTX Multi-Mode Microplate Reader (Winooski, VT, USA), measuring the hydrogel (100 µL) absorbance in the microplate reader for 2 h, at 37 °C, in 30 s intervals, at 350 nm.

### 3.11. Microstructure Characterization of Fibrin/Alginate di-Aldehyde Hydrogels

Before hydrogels were analyzed by SEM microscopy, they were subjected to a supercritical point drying process. For drying the hydrogel, an exchange between water and ethanol was performed in order to properly dry the gels. Hydrogels were left in PBS for 24 h before they were immersed in water/ethanol dilutions of 10%, 20%, 40%, 60%, 80% until reaching a 100% of ethanol solution. Hydrogels were left for 2 h in the 10, 20, 60 and 80% ethanol dilutions, and 24 h in the 40 and 100% ethanol dilutions before supercritical CO_2_ drying. Thar R100W supercritical CO_2_ reactor was used at a temperature of 35 °C and a pressure of 100 bar during 90 min for the drying process. Finally, dried hydrogels were fractured in liquid nitrogen and observed in a Philips XL30 scanning electron microscope.

### 3.12. Rheological Characterization

Rheological tests were performed on the TA Instruments AR-G2 Rheometer (New Castle, DE, USA) using a sand-blasted aluminium plate (25 mm diameter). Samples were gelled a day before the experiment and were left in the incubator at 37 °C to reach equilibrium in the presence of PBS. Gap was fixed for a constant normal force for every sample and the temperature was adjusted to 37 °C. Firstly, the linear viscoelasticity region was determined by means of an oscillatory strain sweep, with amplitude values ranging between 0.01 and 200% at a frequency of 0.3 Hz. For obtaining the G’ and G’’ relationship with frequencies between 0.01 and 20 Hz, a dynamic frequency sweep was run at a fixed strain value, obtained from the previous experiment.

### 3.13. Contraction of FAD Hydrogels

Hydrogels with and without embedded hFBs (20,000 cells/mL) were weighted and photographed at different time points to observe possible changes.

For the contraction studies without cells, 1.5 mL hydrogels was prepared into glass vials for 1 h and detached to a P60 Petri dish, by adding PBS at 37 °C. Then, the hydrogels were extracted, dried, photographed near to a scale bar and weighted in a precision scale balance. Finally, they were soaked again into PBS and incubated at 37 °C until the next measurement. The measurements were performed at 0, 1, 2, 4, 6, 8, 24, 48, and 72 h, and 5, 7, 10, 15, 21, 25 and 30 days. The mass-swelling ratio at each time point is the mean value of 3–6 samples of each weight of the hydrogel, divided by the mass measured of this hydrogel at time zero.

For the contraction studies with cells, 1.5 mL hydrogels was prepared with human primary hFBs at 20,000 cells/mL of gel (into glass vials for 1 h) and detached to a P60 Petri dish, by adding DEMEM at 37 °C. In this case, the hydrogels were soaked into DEMEM, changed every 2 days, and incubated at 37 °C. In each time point, the hydrogels were photographed, analyzed with the imaging processing software ImageJ, and the area-swelling ratio was calculated. This represents the total amount of surface area loss during the incubation. Time points were: 0, 1, 2, 4, 6, 8 h; and 1, 2, 3, 5, 7 and 10 days.

### 3.14. hFB Proliferation Assay

AlamarBlue™ assay was performed in order to study the proliferation of primary hFBs inside the plasma hydrogels. Briefly, fibrin/alginate di-aldehyde hydrogels, with embedded hFBs (80,000 cell/mL) were prepared in 96-well plates with 6 replicates per condition, and measured at 3 different time points: 1, 3 and 7 days. Before the measurement, AlamarBlue reactive was dissolved at 10% in volume into PBS. 100 µL of this AlamarBlue solution was added to the hydrogels and incubated for 3 h at 37 °C in the incubator. After incubation, the supernatants were transferred to a 96-well plate and the fluorescence was measured in the SynergyTM HTX Multi-Mode Microplate Reader (Winooski, VT, USA) (excitation/emission: 570 nm/600 nm). The extracted values, fluorescence in each well, were normalized and represented in a graph, which shows the viability of the hFBs in the hydrogels.

### 3.15. Viability

The viability was measured with a Live/Dead assay, as a complementary experiment to the AlamarBlue assay, in order to study the cells’ viability. 200 µL of hydrogels was prepared inside µ-Slide 8-well glass bottom plates (Ibidi GmbH, Grafelfing, Germany) with embedded hFBs (160,000 cells/mL) and measured at 3 replicates per condition: 48 h and 7 days as time points.

Live/Dead^®^ staining was performed by adding calcein AM and ethidium homodimer from Live/Dead Viability/Cytotoxicity kit (Thermofisher, Frederick, MA, USA) at final concentrations of 0.5 µL/mL and 2 µL/mL in PBS, respectively. 100 µL of this staining was added to the hydrogels and incubated during 40–50 min at 37 °C in the incubator. After this time, the reagent was removed and substituted by PBS. Finally, the hydrogels were observed in a confocal microscope (Leica-SPE (Leica, Wetzlar, Germany)) in the biomedical investigation area in the Gregorio Marañón Hospital. The visualization of the hydrogels in the confocal microscope shows the viability of the cells.

## 4. Conclusions

In this study, we presented a mechanical enhancement of plasma-derived fibrin hydrogels through the incorporation of oxidated alginate (ADA) polymerized using different CaCl_2_ concentrations. This natural anionic polysaccharide was chosen because of its physical-chemical control of hydrogel properties, its biocompatibility, and its biodegradability, in order to improve the mechanical weakness, and rapid degradation and contraction present in plasma-derived fibrin, as well as to use these matrices’ scaffolds in tissue engineering due to their high-water retention capacity and viscoelastic mechanical properties that mimic human living tissues and organs. As alginate di-aldehyde has been recently used to enhanced fibrin hydrogel performance, in this study, its incorporation to platelet-poor plasma is analyzed and characterized. Alginate di-aldehyde seems to inhibit fibrin polymerization when increasing the oxidation percentage above 10%. Since alginate is a natural anionic polysaccharide, the concentration of CaCl_2_ that triggers fibrin polymerization has a strong influence on the final hydrogel microstructure by increasing the fiber diameter and mechanical properties. Finally, human primary fibroblasts were cultured inside the hydrogels, showing that for higher oxidation degrees, the cell viability is reduced. In this sense, larger CaCl_2_ concentrations seem to enhance cell spreading and proliferation. These results indicate the potential of using these types of hydrogels in tissue engineering for the production of autologous skin grafts, skeletal muscle regeneration and bone tissue reparation.

## Figures and Tables

**Figure 1 ijms-23-04296-f001:**
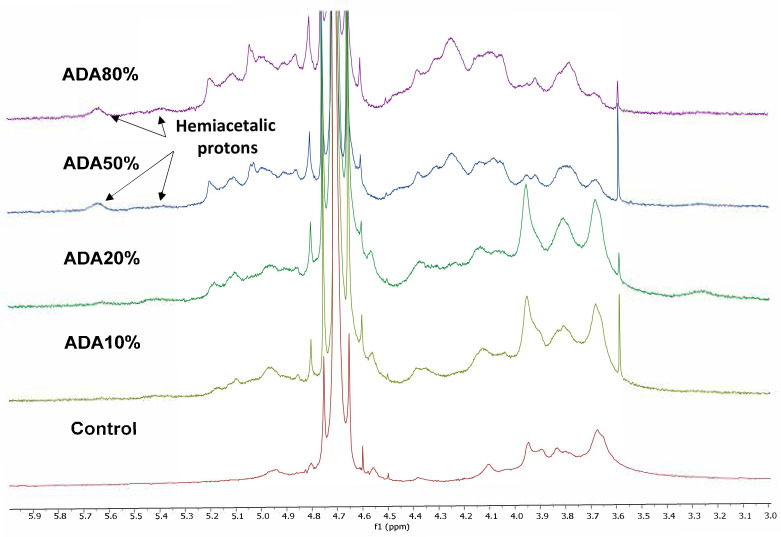
H^1^-NMR spectra of different samples of sodium alginate (control) and alginate-di-aldehyde with different oxidation degrees.

**Figure 2 ijms-23-04296-f002:**
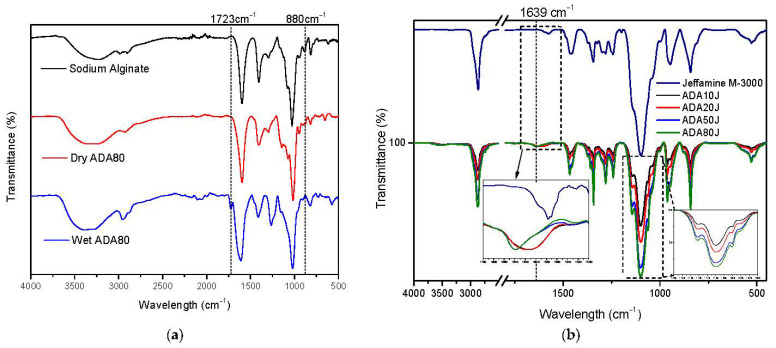
(**a**) FTIR spectra of sodium alginate, dry ADA80 sample and wet ADA80 sample, from top to bottom. (**b**) FTIR spectra comparison of ADA 10, 20, 50 and 80 that reacted with Jeffamine M-3085.

**Figure 3 ijms-23-04296-f003:**
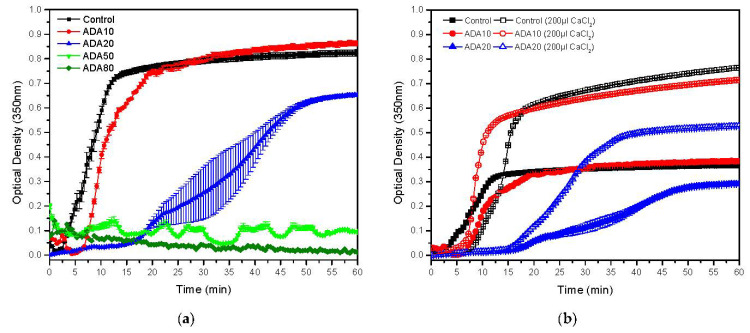
Gelation kinetics for: (**a**) fibrin hydrogels with alginate at different oxidation degrees and (**b**) fibrin hydrogels and alginate at different oxidation degrees gelled with different amounts of CaCl_2_ using UV spectrometry at 350 nm. Data shown as mean ± SD, *n* = 3.

**Figure 4 ijms-23-04296-f004:**
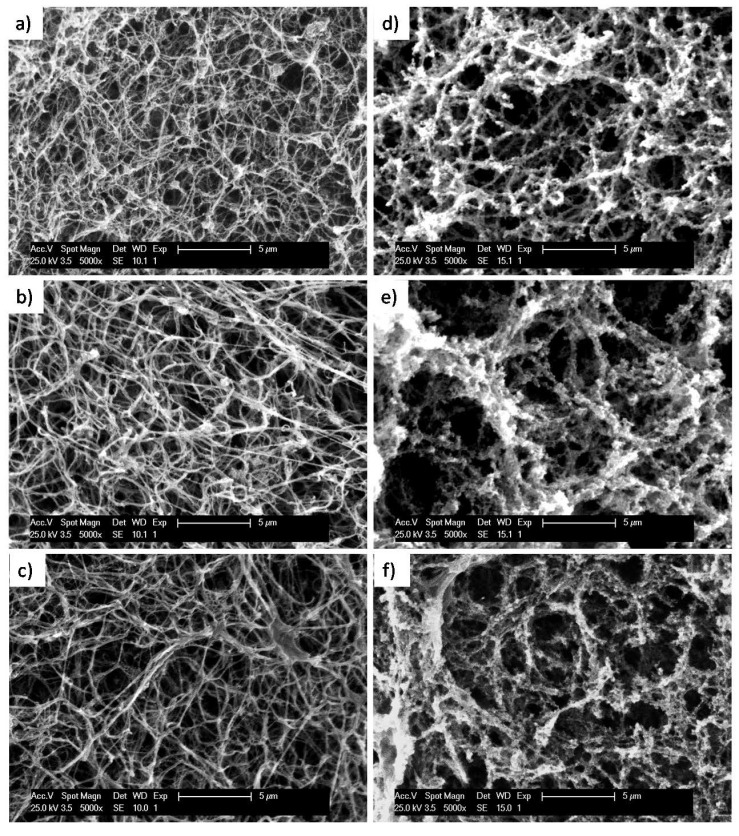
Micrographs of hydrogel microstructures. On the left, fibrin (**a**), fibrin/ADA10 (**b**) and fibrin/ADA20 (**c**) gel microstructure for 80 µL CaCl_2_ per mL of gel. On the right, fibrin (**d**), fibrin/ADA10 (**e**) and fibrin/ADA20 (**f**) gel microstructure for 200 µL CaCl_2_ per mL of gel.

**Figure 5 ijms-23-04296-f005:**
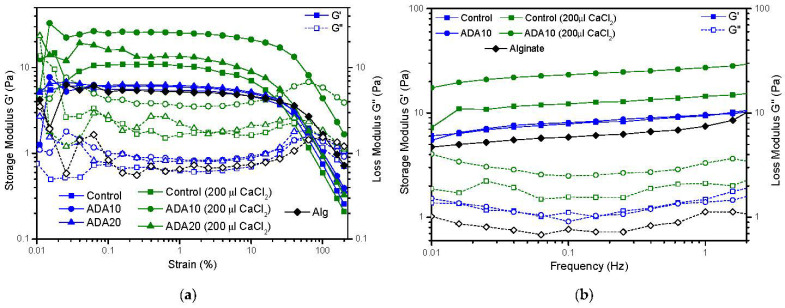
(**a**) Strain sweep test of different plasma-derived fibrin hydrogels after 24 h in PBS at 37 °C. (**b**) Frequency sweep test of representative differences between different plasma-derived fibrin hydrogels after 24 h in PBS at 37 °C.

**Figure 6 ijms-23-04296-f006:**
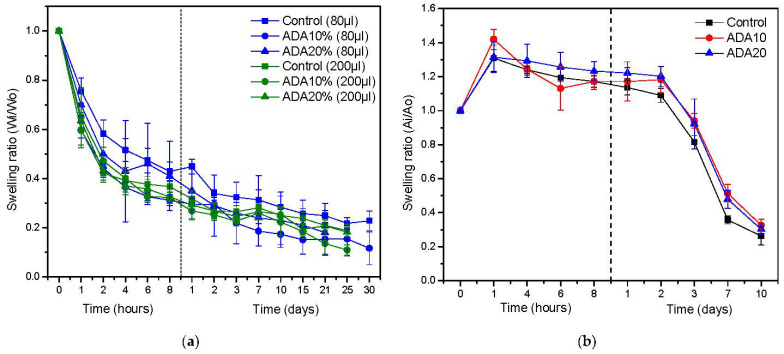
Plasma-derived contraction of fibrin hydrogels measured as: (**a**) mass loss in PBS at 37 °C modified with different oxidized alginate and amounts of CaCl_2_; (**b**) area reduction when hFBs were embedded inside the hydrogels. Data shown as mean ± SD, *n* = 3.

**Figure 7 ijms-23-04296-f007:**
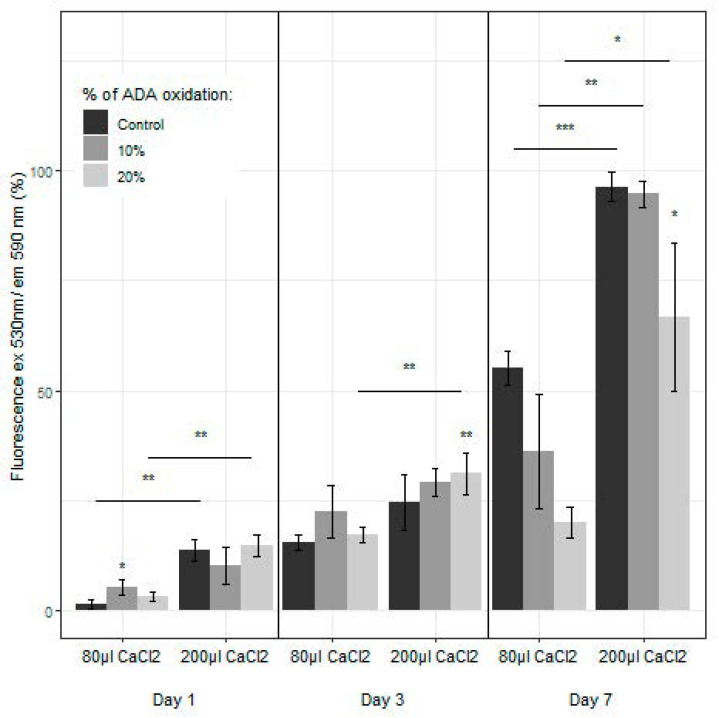
Fibroblast proliferation inside the plasma-derived hydrogels measured through the AlamarBlue™ assay, with 2 mg/mL of modified alginate at 10% and 20% oxidation and different concentrations of CaCl_2_ (80 µL and 200 µL), at different time points (1, 3 and 7 days). (* *p* < 0.05, ** *p* < 0.01, *** *p* < 0.001).

**Figure 8 ijms-23-04296-f008:**
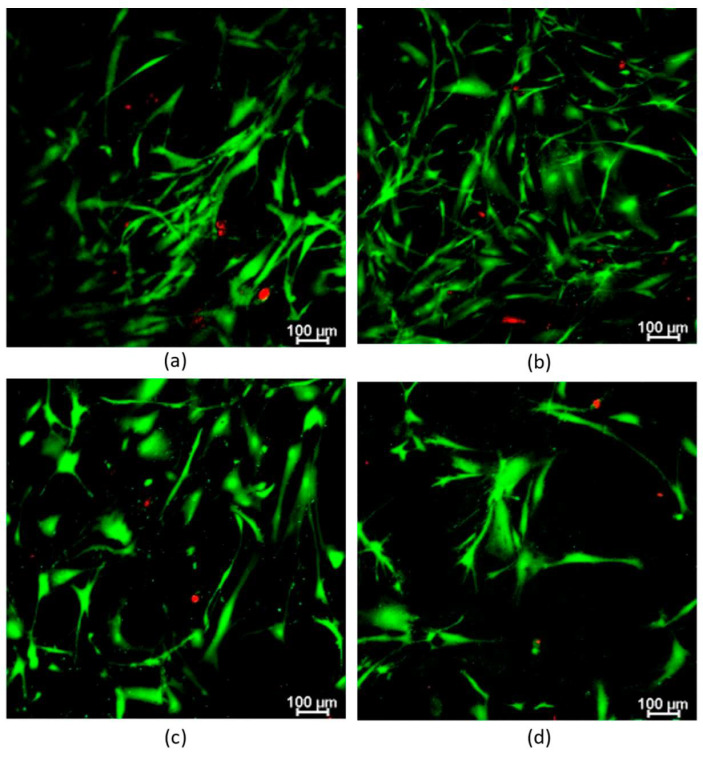
Representative images of fibroblast proliferation inside the plasma-derived hydrogels at 1.2 mg/mL final fibrin concentration measured through Live/Dead^®^ with (**a**) 2 mg/mL of 20% oxidized alginate and 200 µL of CaCl_2_ used to allow the gelation reaction, after 7 days of incubation; (**b**) plasma-derived hydrogel control and 200 µL of CaCl_2_, after 7 days; (**c**) 2 mg/mL of 10% of oxidized alginate and 80 µL of CaCl_2_, after 7 days; and (**d**) 2 mg/mL of 10% of oxidized alginate and 80 µL of CaCl_2_, after 48 h.

**Table 1 ijms-23-04296-t001:** Oxidation percentage, weight-average molar mass, polydispersity index (PI) and the maximum amount of each oxidized alginate sample that allows for fibrin hydrogel formation.

Theoretical Oxidation (%)	Oxidation (%) ^1^	*M*_W_ (GPC)	PI	Maximum Concentration in Final Hydrogel (mg/mL)
0	0	467,000	2.3	-
10	7.7 ± 0.5	178,000	1.9	3
20	17 ± 2.6	69,000	1.6	3
50	46.4 ± 2.5	32,000	2.9	0
80	74.1 ± 3.1	33,000	2.5	0

^1^ Data shown as a mean ± SD, *n* = 3.

**Table 2 ijms-23-04296-t002:** Oxidation reaction of sodium alginate with different molar ratios of sodium metaperiodate.

Theoretical Oxidation (%)	NaIO_4_ (mmol)	Sample Name
0	0	Sodium Alginate
10	0.5	ADA10
20	1	ADA20
50	2.5	ADA50
80	4	ADA80

## Data Availability

Due to the size of the raw files, datasets are only available upon request.

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
