# Peer review of "Preparation and Characterization of Plasma-Derived Fibrin Hydrogels Modified by Alginate di-Aldehyde"

_ijms, 2022, doi:10.3390/ijms23084296_

Round 1

Reviewer 1 Report

The manuscript by Raul et al. on "Analysis and characterization of plasma-derived fibrin hydrogels modified by alginate di-aldehyde" describe the preparation and characterization of ADA-Fibrin hydrogel formulations. Overall, the study was well done and clearly presented. Authors may clarify the following points for further consideration.

Major comments

Authors claim that the the quantity of CaCl2 influences the spread of the hFBs. This was based on fluorescence microscope images in Figure 8. There are two issues here. (a) The claim based on mere microscope imaging is debatable. Authors may need to prove this claim with further experimentation with quantitative means. (b) Perhaps, authors may know that cell adhesion and spreading would be influenced by the availability and distribution of the focal adhesion points on the substrate. Instead of attributing the changes to CaCl2 directly, authors may highlight the morphological features, surface chemistry changes, topography changes, serum protein adsorption, etc. which directly influence the cell adhesion and spreading.

Minor comments

  1. "Analysis" and "characterization" terms in the title wouldn't mean the same!
  2. Key peak positions may be annotated in Figure 1.
  3. Spacing may be adjusted to distinguish different panels in Figure 4.
  4. Is it possible to convert data into "cell viability%" in Figure 7?

Reviewer 2 Report

The authors present oxidized alginate (ADA) fibrin hydrogels and assess the influence of different degrees of oxidation of alginate as well as Ca2+ concentration on the physico-chemical properties of the hydrogels. In addition, the authors characterize cytocompatibility of the hydrogels using human primary fibroblasts, showing cell attachment inside the hydrogels, as well as survival and proliferation in the presented hydrogels. The authors identify a critical degreeOfOxidation of ADA, where results suggest that cell proliferation is mitigated.

However, the manuscript lacks partly in a specific language, as well as sometimes features wrong terminology which hampers readability. The methods section lacks critical details required for reproducibility of the results. In addition, the novelty of the work is not clear, as very recently, a similar study on oxidized alginate in combination with fibrin has been released: https://doi.org/10.1016/j.msec.2022.112695. Further, the paper lacks significant discussion with previously published works in fibrin- and ADA-based hydrogels (What is new in the presented work? What has been done before? How are the presented materials advantageous in light of previous hydrogels using similar materials?). Such discussion is vital to increase the potential impact of the presented study, and to clarify the novelty of the presented work.

The manuscript hence requires major revision prior to be considered for publication.

In detail:

P1, L15. Specificity. The authors refer to “excellent biological properties”. What does this mean? Specify.

P1, L18. English. Please change “reparation” to read “repair”.

P1, L18. Specificity. The authors refer to “mechanical weakness”. What do the authors refer to specifically? Low stiffness in comparison to native soft muscle tissue? Please specify the statement in light of a specific application to which the statement refers to.

P1, L22. The authors refer to “ADA oxidation” inhibits (..) polymerization. Please specify. Do the authors mean “degree of oxidation of ADA”? or “Alginate oxidation”? The authors must make sure that the language used avoids ambiguity and is specific. Also, the authors refer to “which could imply a fiber diameter increment”. It is suggested that this statement, which is later on described as a result in the paper, is transferred to the discussion. Or, the authors rewrite to read “which we associate with an fiber diameter increment”, if the authors see this as an clear result from their study.

P1, L25. The authors refer to “80µl and 200µl” as concentrations. However, those are measures of volume. The nomenclature used must be scientific and correct. Please rewrite this statement.
Also, please specify what the authors mean by “contraction”. Did the authors measure contraction as a mean of displacement upon force inside the hydrogel? Please specify. Otherwise, the statement is unclear.

P1, L30. The authors highlight the potential cytotoxic effect of highly oxidized alginate on human primary fibroblasts. Previous researchers assessed the event of reactive oxygen species (ROS) as a result from using oxidized alginate with fibroblasts, leading to decreases in viability (https://www.mdpi.com/1422-0067/22/5/2358). However, the authors do not discuss this study. Please assess this paper in light of your results, which may be the reason for decreased viability with an increasing degree of oxidation. Such discussion would increase the comprehensiveness and visibility of the presented study.

P4, Figure 1. The figure graphs seem to be cropped at the top, which might be due to the high peak at ~4.7 ppm. The authors should transfer Figure 1 to the supplementary information and correct this issue. Please display the full data. In addition, please highlight the regions in the 1HNMR spectrum which indicate successful oxidation of alginate (e.g., representing aldehyde groups, as well as guluronic acid groups like correctly indicated at P3, L111).

P4, L149: Please change “consonance” to read “correlating”. The authors should provide the FTIR spectrum of monojeffamine as reference in figure 2b. Schiff base formation with jeffamine should be visible by a peak shift from the amine peaks of jeffamine in comparison to imine-bond formation between amine peaks of jeffamine and the oxidized alginate.

P7, Figure 4. The authors provide interesting results which indicate jamming of microfibers of fibrin in the presence of ADA of a higher degree of oxidation. However, the impact on the hydrogel mechanical properties is not clear from Figure 5 (potential softening from 20 to ~10 Pa G’, Figure 5b), which would be very interesting to see. Please provide AFM data of the hydrogels if accessible, of compression, texting of hydrogel cylinders, to assess the mechanical properties of Control and ADA10 and ADA20 gels in greater depth. This would increase the quality and comprehensiveness of the work.

P8, Figure 5. The authors show mechanical data of their hydrogels. The resulting hydrogels appear to be very soft (~10 – 40 Pa), similarly to e.g. brain tissue. However, the authors show fibroblast spreading and survival inside the hydrogels. Please provide relevant discussion on how fibroblasts could adhere and subject forces on the hydrogel, required for cell spreading, even though hydrogel modulus is very low, as similar results for spread fibroblasts have been observed for oxidized alginate hydrogels in the range of kPa before (e.g. https://doi.org/10.1002/mabi.201900245).

P9, Figure 6. The authors show a swelling experiment. However, the authors must provide information on: Number of replicates (n=?) data shown (mean/median?) and error bar meaning (Standard deviation? S.E.M.?) This accounts for all data shown in the manuscript.

P11, L336. The authors must provide information from which suppliers the reagents in their study were purchased (missing. E.g. L339. Sodium metaperiodate, alginate, etc.)

P11, L338. The authors describe that the synthesis of ADA was performed according to previous work. If stated so, please cite the previous work from which the authors adapted their synthesis protocol. Reference missing. Also, it must be clear which kind of alginate the authors used (%guluronic/mannuronic acid content? Molecular weight? vendor Etc.)

P11, L341. The authors refer to “mol.%”. Please specify the amount of sodium metaperiodate that was used. Please provide a table clearly stating the amount of sodium metaperiodate (or the quantity of substance [mmol]) that was added. This would increase the readability and clear reproducibility of the synthesis performed in the study.

P12, L372. The authors describe an interesting way of reaction of ADA with “Jeffamine®”. Please provide information on the dialysis performed (MWCO? Tubings used?). If not citing other sources, the authors must shortly iterate on the way of function of their assay. As FTIR-ATR is not easily allowed to quantify from the peaks acquired, the methodology must be described in greater detail to clarify the analysis.

P13, L401. Please define what “DEMEN” is. Please change “ADA were” to read “ADA was”. English.
The authors must rewrite section 3.7. to feature clearly, in a step-by-step manner, how the hydrogels were prepared. This will dramatically increase the reproducibility of the manuscript as well as the quality of the manuscript.

P13, L444. The authors refer to “Contraction” studies, while basically performing measurements of mass change or a “Swelling and degradation” study. Contraction is misleading as it refers to shrinkage, or better to a deformation of the gels upon force. Please change the terminology to “Swelling and degradation” or “hydrogel shrinkage upon degradation” to avoid ambiguity.

P14, L476. Terminology. A viability study is not suitable to assess cell morphology but to assess viability. The authors can clarify that via calceinAM, cells were imaged using a fluorescent microscope, and cell spreading inside the hydrogels was observed. However, the authors should rewrite this section as live dead staining is not perfectly suitable to assess morphology or proliferation. Proliferation should be assessed, e.g., via the alamarblue assay, cell counting over time, or Ki-67 staining. Morphology can be assessed via, e.g., F-Actin cytoskeleton staining.

The methods section lacks a clear section stating cell culture and maintenance. It is not clear where primary human fibroblasts were obtained from (purchased? Primary isolated?).

The lack of detail in the methods section accounts for all subsections. The authors did not introduce acronyms such as “FTIR”. The ethanol series used for water exchange in SEM analysis was not specified. Etc. Please rewrite and provide all necessary information in the manuscript.
